# communications

# earth & environment

# DNA sequencing, microbial indicators, and the discovery of buried kimberlites

Rachel L. Simister[1,2], Bianca P. Iulianella Phillips[2,3], Andrew P. Wickham[2,3], Erika M. Cayer[2,3], Craig J. R. Hart[2,3], Peter A. Winterburn[2,3,4] & Sean A. Crowe [1,2✉]

Population growth and technological advancements are placing growing demand on mineral resources. New and innovative exploration technologies that improve detection of deeply buried mineralization and host rocks are required to meet these demands. Here we used diamondiferous kimberlite ore bodies as a test case and show that DNA amplicon sequencing of soil microbial communities resolves anomalies in microbial community composition and structure that reflect the surface expression of kimberlites buried under 10 s of meters of overburden. Indicator species derived from laboratory amendment experiments were employed in an exploration survey in which the species distributions effectively delineated the surface expression of buried kimberlites. Additional indicator species derived directly from field observations improved the blind discovery of kimberlites buried beneath similar overburden types. Application of DNA sequence-based analyses of soil microbial communities to mineral deposit exploration provides a powerful illustration of how genomics technologies can be leveraged in the discovery of critical new resources.

---

[1] Department of Microbiology & Immunology, University of British Columbia, Vancouver, BC V6T 1Z3, Canada. [2] Department of Earth, Ocean and Atmospheric Sciences, University of British Columbia, Vancouver, BC V6T 1Z4, Canada. [3] MDRU-Mineral Deposit Research Unit, Department of Earth, Ocean and Atmospheric Sciences, University of British Columbia, Vancouver, BC V6T 1Z4, Canada. [4]Deceased: Peter A. Winterburn. ✉email: sean.crowe@ubc.ca

Microorganisms operate together with geological processes to drive biogeochemical cycles that shape Earth's surface chemistry and climate through time[1]. They interact with minerals at the nano- to microscales[2], and these interactions give rise to emergent properties across the multiple scales that characterize the biosphere[3]. Through billions of years of evolution, microorganisms have honed their ability to sense and interact with their surrounding environments and, in particular, to respond to the availability of mineral nutrients and substrates. Through their metabolism, microorganisms affect the distribution of minerals at Earth's surface—in extreme cases this can lead to the formation of mineral resources[4–6]. Microbial community compositions and structures are thus sensitive reflections of their habitats[7], and analyses of microbial communities can provide a wealth of information on their surrounding environments.

High-throughput sequencing technologies now allow us to analyze microbial communities and leverage microbial sensing to interrogate the environment with unprecedented sensitivity and resolution[8–10]. Sequence-based microbial community analyses, for example, have been used to detect organic and inorganic contaminants in groundwater at the watershed scale[8]. They have also been used as pathfinders in petroleum exploration[11]. More broadly, microbial communities are known to respond to a wide range of physical–chemical properties, including pH[12], salinity[13], temperature[14,15], light intensities[16], and mineral micronutrients[17], among others. Historically, the application of microbiology in the natural resource sector has mostly been limited to mineral processing, predominantly bioleaching of sulfide ores[18]. For example, acidophilic microorganisms, such as *Acidithiobacillus ferrooxidans*[19], which are abundant in natural environments associated with pyritic ore bodies, coal deposits[20], and their associated acid mine drainages[21], have been harnessed at commercial scale to extract copper and gold from sulfide ores for decades[22]. In contrast, we have mostly overlooked the potential power of microbial communities to enable resource discovery in the natural environment and are only just beginning to harness the capacity of environmental microbial communities as environmental resource indicators. At the same time, demand for mineral resources is increasing, existing mineral deposits are being mined out, and the frequency of new deposit discovery is declining[23–25]. Furthermore, the contemporary toolkit for mineral deposit exploration consists of a suite of geophysical and geochemical approaches that often fail to appropriately delineate concealed mineralization in the areas with thick and complex overburden that likely host the vast majority of currently unknown mineral resources[23]. Given that new demand for mineral resources must be increasingly met through discovery and development of deeply concealed deposits[26–28] microbial communities could play an important role if they respond to subsurface mineralization.

Innovation and the development of new tools and techniques are needed to improve our ability to find mineral resources. It has been known for more than half a century that vegetation responds to subsurface geologic features through the influence of bedrock geology on the physical and chemical properties of surface soils[29,30]. The link between vegetation patterns and bedrock geology prompted the early use of biological surveys in mineral deposit exploration[31–33]. Vegetation patterns, however, are confounded by many variables[34,35], and thus rarely offer clear indications of buried mineral deposits. Use of biological surveys in exploration has been extended to soil microbial communities[36–40], but the complexity of these communities is intractable through the approaches of classical microbiology, while early-generation molecular approaches lacked throughput[41–43]. Now, however, even the most complex microbial communities, like those found in soils, can be resolved through semi-quantitative to quantitative sequence-based analyses[12,44,45]. Given that every gram of soil contains thousands of microbial taxa[46,47], each housing hundreds to thousands of genes sensing and interacting with the surrounding soil environment[48,49], the power of this approach to identify anomalies in soils is unprecedented.

In this study, we tested the potential for soil microbial communities in sub-arctic tundra to respond to and thus indicate ore materials and buried mineralization. Tundra soils and soils derived from glacial tills are geographically expansive, overlie diverse bedrock lithologies, and indeed likely conceal a wealth of buried mineral resources that remain undiscovered. Kimberlites, in particular, are conspicuous examples of mineral resources commonly concealed by glacially derived tundra soils. Kimberlites are variably serpentinized, high-Mg, ultramafic rocks that are host ores of natural gem and industrial quality diamonds and are increasingly considered important source materials for atmospheric carbon capture and storage technologies[50,51]. We conducted a combination of both incubation experiments, in which these soils were amended with kimberlite rock pulp, and exploration-style soil surveys across buried diamondiferous kimberlite pipes. Microbial community compositions in these experiments and surveys were determined through amplicon sequencing of the 16 S rRNA gene. Bioinformatic analyses were used to establish a suite of microbial indicator species. Collectively our results show that mineral deposits buried under 10 s of meters of soil and unconsolidated surficial materials can be located at the surface through microbial community profiling using high-throughput DNA amplicon sequencing.

## Results and discussion

**Microbial community responses to kimberlite materials**. Incubation experiments reveal that microbial community compositions and structures respond directly to amendments with ground rock from diamondiferous kimberlites. We amended tundra-derived soils with pulverized (80% passing 10 mesh (2 mm)) kimberlite (5% w/w) and analyzed the response of soil microbial communities through amplicon sequencing of the small subunit (16 S) ribosomal rRNA gene. At 5%, amendment with kimberlite had a modest effect on overall soil chemical composition (Supplementary Data 1). Baseline soils had microbial community compositions comprised predominantly of 6 phyla— Proteobacteria, Actinobacteria, Bacteroidetes, Acidobacteria, Chloroflexi, and the WPS-2 candidate phylum (Fig. 1a, b and Supplementary Fig. 1). The soils also contained appreciable, but lesser, proportions of Verrucomicrobia, Planctomycetes, and Gemmatimonadetes (Fig. 1a). Such community compositions are typical of both tundra soils[52–54] and a broad suite of geographically disparate soils, more generally[45]. We found that over a period of 85 days, the microbial community composition and structure in soils amended with 5% w/w kimberlite, diverged from the baseline with pronounced changes observed at the phylum level, including increases in the abundances of Proteobacteria and Bacteroidetes from 46 to 68% and 6 to 16%, respectively, in response to amendment (Fig. 1a, b and Supplementary Fig. 1). Four phyla (Chloroflexi, Acidobacteria, Actinobacteria, and WSP-2), on the other hand, decreased from 6%, 5%, 19%, 7% in the baseline to 1%, 2%, 8%, 1%, following amendment, respectively (Fig. 1a, b and Supplementary Fig. 1). Experimental results thus reveal that the addition of kimberlite (5% w/w) to tundra-derived soils causes strong shifts in microbial community composition that are easily resolved through amplicon sequencing of the 16 S rRNA gene.

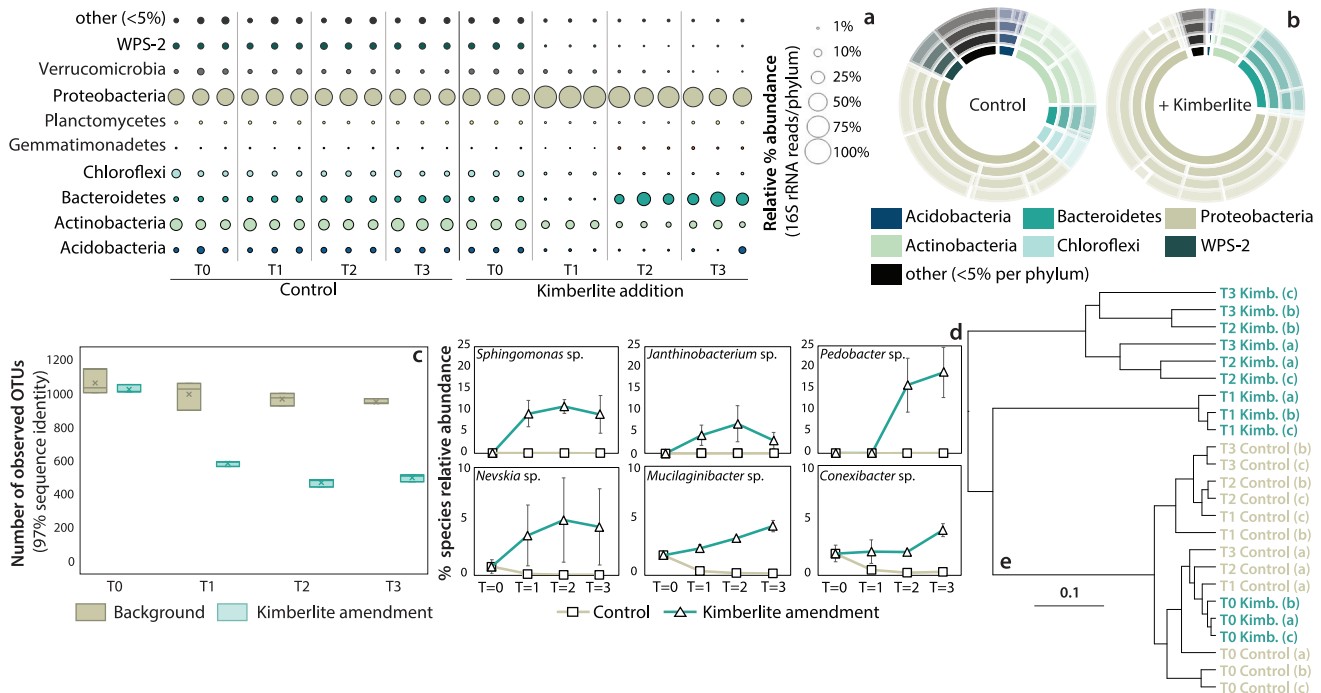

**Fig. 1 Soil microbial community composition, diversity, and indicator species for the kimberlite amendment experiment. a** Distribution of 16 S rRNA gene reads per phylum for each sample. The number of reads per phylum is calculated as a percentage of the total reads for each sample. The "other" grouping represents phyla that when summed contributed (on average across all samples) <5% of the total number of reads per sample. **b** A sunburst chart shows the average total relative abundance of bacterial and archaeal communities in control soils and kimberlite-amended soils. Rings are ordered as follows from inner to outer: Phyla, Classes, Orders, Families, and Genera. **c** Number of observed operational taxonomic units (OTUs; 97% sequence similarity) per sample at each timepoint, colored by sample treatment (from data that has been rarefied to 16365 sequences per sample). Median values are indicated by the solid line within each box, and the box extends to upper and lower quartile values. **d** Examples of OTU (species-level) changes across treatments, over time. Error bars represent standard deviation. **e** Hierarchical relationships amongst control and kimberlite treated soils based on Euclidean distance of 16S-OTU abundances. The hierarchical relationships between soil samples were obtained using the unweighted pair group method with the arithmetic mean (UPGMA) clustering algorithm. Node labels indicate the timepoint/treatment.

Amendment with kimberlite material was sufficient to cause appreciable changes to microbial community structure and a decline in diversity at the species level (97% sequence identity in the 16 S rRNA gene), relative to the baseline. Whereas metrics for microbial diversity can be hard to interpret, some potentially informative patterns emerge from our data. Diversity indices like Chao1, for example, show a decrease in species richness from $1610 \pm 70$ in the baseline soils to $830 \pm 60$ following amendment (Supplementary Data 2). This decline in species richness is also supported by a decline in the number of observed OTUs relative to baseline, which decrease by 48% (on average $990 \pm 10$ in control samples, and $520 \pm 10$ in kimberlite-amended soils) (Fig. 1c and Supplementary Data 2). Reduction in species richness in response to amendment is likely due both to selective growth of some taxa and the death and decay of others. This is consistent with limited net community growth overall during the incubations, based on qPCR assays of 16 S rRNA gene abundance (Supplementary Fig. 2). Differences in microbial community composition and structure between baseline and amended soils were evaluated through hierarchical clustering analysis (Fig. 1e). All baseline soils clustered tightly, exhibiting both similar bacterial diversities and microbial community compositions, whereas amended soils grouped separately. This confirms that kimberlite amendment induced clear and reproducible shifts in microbial community compositions and structures, demonstrating that major features of soil microbial communities are sensitive to the presence of kimberlite materials on timescales of several weeks.

Beyond high-level changes in taxonomic composition and structure, many individual species responded to kimberlite amendment. An indicator species analysis (Supplementary Data 3a, b) revealed a total of 375 species that responded significantly (Linear Discriminant analysis (LDA) threshold score >2) to kimberlite amendment, and thus qualify as indicators for kimberlite material. Of these, 65 species (17%) increased in relative abundance over the 85-day incubation period, whereas 310 species (83%) decreased in relative abundance, with respect to the baseline (Fig. 1d and Supplementary Data 3a, b). Notable examples of species that increased in abundance include *Sphingomonas* sp., *Janthinobacterium* sp., and *Pedobacter* sp., whereas species that decreased include *Nevskia* sp., *Mucilaginibacter* and *Conexibacter* sp., (Fig. 1d and Supplementary Data 3a, b). Comparisons of the indicator species to a database of microbial functions[55] imply that the indicator species are associated with a wide range of metabolic potentials that are common and widely distributed in soil microbial communities (Supplementary Data 3a, b). It is important to point out, however, that inferences of function from taxonomy are prone to error that arises from strong differences in metabolic potential across closely related taxa. Collectively, the 65 species that increased in abundance following amendment made up 60% of the total community following incubation, relative to 0.6% in the baseline. Following incubation, these 65 species exhibited a mean of 1% and median of 0.2% in amended soils, versus 0.01% and 0% in the baseline, respectively. Similarly, the 310 species that decreased in abundance following amendment made up 8% and 74% of the total community in amended and baseline soils, respectively. These

species exhibited a mean of 0.027% and median of 0.007%, versus 0.24% and 0.065% in amended versus baseline soils, respectively, following incubation. These results thus demonstrate that amendment with kimberlite induces a fundamental reorganization of generally low-abundance microbial community members with the overall effect of entirely changing the microbial community composition and structure in a matter of a few weeks. Furthermore, amendment with kimberlite selects for ingrowth of members of the rare biosphere, that were mostly undetectable prior to incubation, to abundances of several % (e.g., *Janthinobacterium* sp), whereas other members dropped from several percent, to obscurely low abundances (e.g., *Nevskia* sp) (Fig. 1d and Supplementary Data 3a, b). This demonstrates that microbial communities are exquisitely sensitive and responsive to subtle variations in the mineral composition of soils, with strong potential for detection of this variation in the environment.

**Microbial community profiling over buried mineralization.** Tundra soils analyzed over buried diamondiferous kimberlite mineralization in northern Canada (Northwest Territories) reveal largely homogenous microbial community compositions, but also differences in diversity that are spatially related to the surface expression of the underlying kimberlite (Fig. 2b). The B-horizons of soils that developed on up to 20-m-thick glacial tills were sampled in a grid pattern across the surface expression of a kimberlite body that has been well-defined by drilling (kimberlite DO-18) (Supplementary Fig. 3a). The surface materials are dominated by till in the northern section of the sampled area and till, glaciolacustrine clay, glaciofluvial silt, sand, gravel, and organic deposits in the south. Most soil microbial community members belong to the Proteobacteria, Acidobacteria, Verrucomicrobia, and Actinobacteria phyla (Fig. 2a), which is comparable to the dominant phyla in the soils used in our incubation experiments, soils from other tundra environments[52–54], and soils globally[46,48]. At higher taxonomic resolution (species level, 97% sequence identity), the soils comprised a number of conspicuous and abundant taxa, including an uncultured *Gemmatimonadaceae* sp.; an unknown Chloroflexi belonging to the class AD3; a Candidatus udaeobacter species; a *Neviskia* sp.; and another Candidatus udaeobacter species, all of which were greater than 2% on average, and commonly found in soils (Supplementary Data 4). Notably, the Candidatus udaeobacter are the most abundant organisms in soils, globally[56,57]. Whereas microbial community compositions are known to vary appreciably at local, regional, and global scales and in response to prevailing environmental conditions, a relative homogeneity of the dominant microbial community members was observed across the sampling grid (Supplementary Data 4). The number of species observed ranges from 497 to 2025 with a mean of 1400 ± 300, and estimates of total species richness (Chao1) range from 737 to 3306 with a mean of 2300 ± 580, implying that these soils have diversity typical of other soils, which can range from 100's to thousands of observed species per sample (Supplementary Data 5 and 6)[46,58]. Furthermore, estimates that also consider community evenness (Inverse Simpson) imply that species abundances are not evenly distributed in these soils (inverse Simpson indexes range from 16 to 131, with a mean of 72 ± 29) (Supplementary Data 5 and 6). When these soils are grouped according to their spatial relationships to the surface expression of the kimberlite body, we find that microbial species richness in soils directly overlying the kimberlite is, on average, 29% lower than that in the background soils (average chao1 index of 1840 ± 80 above surface projection of the kimberlite and 2600 ± 100 above background), which are geographically removed from the underlying kimberlite (Fig. 2b). Whereas differences in community structure reflect proximity to

buried kimberlite, high-level community compositions do not appear sensitive to buried kimberlite mineralization, and the abundances of the major microbial phyla are similar across the entire sampling grid (Fig. 2a).

Differences in microbial community compositions of soils situated directly above the surface expression of kimberlite, and those of background soils can be observed through statistical analyses conducted at the species level. Hierarchical clustering analyses demonstrate that soils situated above buried mineralization have microbial communities that are more similar to each other than they are to the background soil communities. Several clusters had more than 50% of soils located above the surface expression of the kimberlite (clusters 1, 4, and 5), whereas some clusters only contained background soils (clusters 2, 3, 6, and 7) (Fig. 2c). This implies that though high-level differences in phyla, like those observed in the incubation experiments, may not be expressed in natural settings, there are more subtle differences in community composition that are resolvable through more nuanced analyses.

Species-level fingerprints identified through indicator species analyses successfully resolve soils that overlie buried kimberlite. Of the 65 indicator species identified through the incubation experiments, 59 were present in soils surveyed around the DO-18 deposit. 19 of these indicators, furthermore, were appropriately enriched in soils overlying the buried mineralization, relative to the background, and thus effectively resolve the surface expression of the kimberlite (Fig. 2e). We also conducted an indicator species analysis by comparing microbial communities overlying the surface expression of the deposit to those from background soils and this yielded a further 59 indicator species, 2 of which were the same as those identified through incubations (Supplementary Data 3). Albeit small (3%), the overlap in indicator species between the incubation soils and the soils from DO-18 suggests that collections of indicator species can be more broadly extensible, at least across similar types of mineralization, and in comparable soil terrains. Combining the field-based indicator species with those from the incubation experiments yields a collective set of 78 indicators (with a summed average relative abundance of 10 ± 7%, Supplementary Data 7) and generation of anomaly maps with this combined indicator set very effectively resolves the underlying kimberlite (Fig. 2f). For comparison, we have also employed commonly used geochemical kimberlite pathfinder elements including Cr, Ni, Mg, and Nb (Fig. 2g). These pathfinder elements display an anomaly pattern that indicates glacial transport of kimberlite material away from the bedrock source and yields responses that are geographically less precise and quantitatively less pronounced (Fig. 2d) than the microbial indicators. Comparing the response ratios for geochemical and microbiological indicators it becomes immediately evident that DNA sequence-based microbial community profiling much more effectively resolves the location of buried kimberlite mineralization than the geochemical data and suggests that amplicon-based microbial community profiling can provide a robust and surgical mineral exploration tool.

**Application of microbial community profiling to blind discovery of buried mineralization.** As a proof-of-concept, we used microbial indicators derived from our incubation experiments and analyses of DO-18 soils to resolve kimberlite mineralization at another location (Kelvin) in the Northwest Territories (Supplementary Fig. 4). The Kelvin kimberlite is overlain by approximately 4 m of glacial till, and up to 150 m of bedrock covers the underlying kimberlite deposit (Supplementary Fig. 3b, c). Soils here are composed of poorly sorted clay, silt, sand, gravel, as well as dispersed boulders (diamicton). Microbial community compositions at

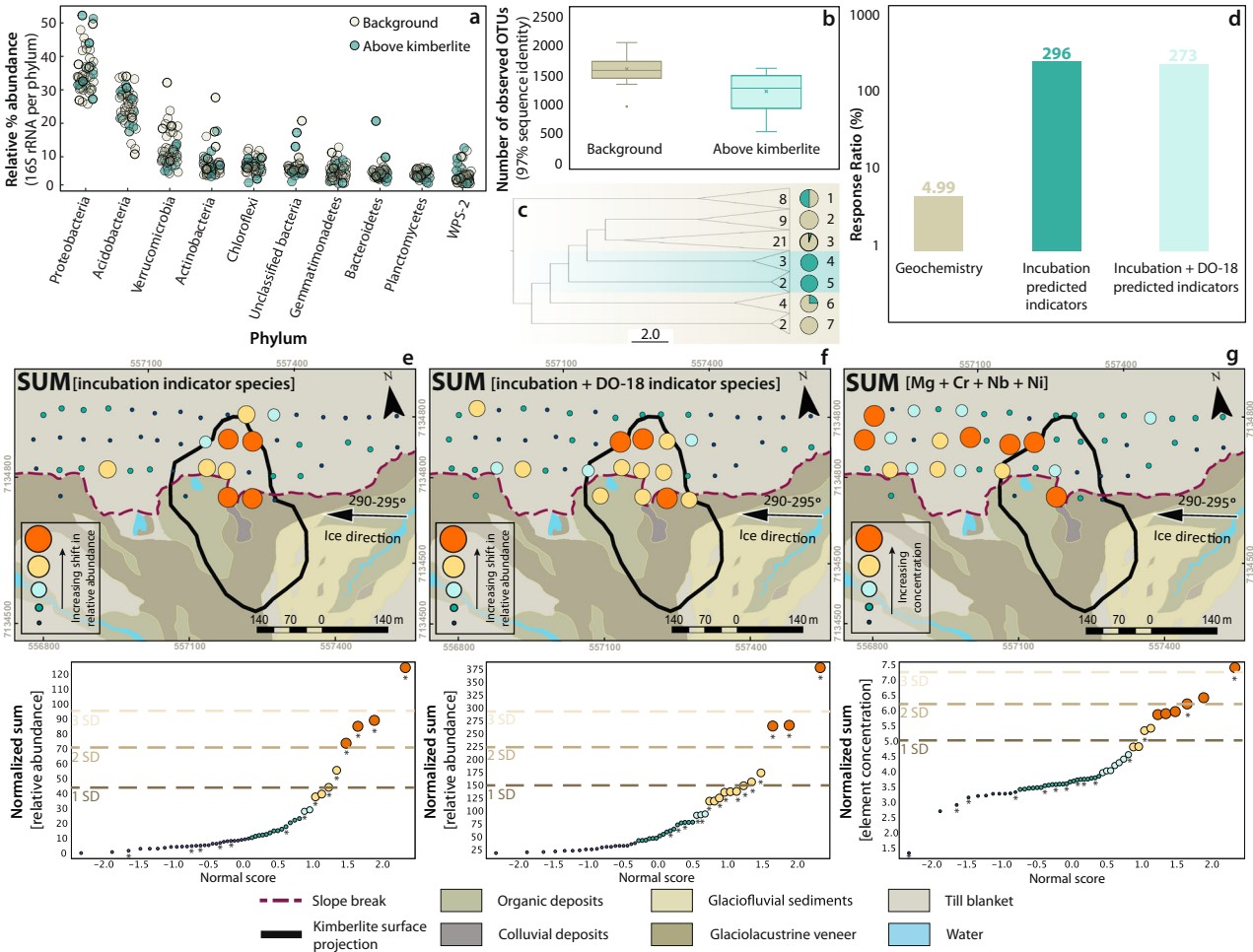

**Fig. 2 Soil microbial community composition, diversity, and indicator species for the DO-18 kimberlite. a** Distribution of 16 S rRNA gene reads per phylum for each sample at the DO-18 kimberlite. The number of reads per phylum is calculated as a percentage of the total reads for each sample. **b** Box plots show the number of observed operational taxonomic units (OTUs; 97% sequence similarity) across background soils and soils above the DO-18 kimberlite (from data that has been rarefied to 16365 sequences per sample). Median values are indicated by the solid line within each box, and the box extends to upper and lower quartile values. Stars indicate the mean. Error bars represent standard deviation. **c** Hierarchical relationships amongst soil samples are based on Euclidean distance of 16S-OTU abundances. The hierarchical relationships between soil samples were obtained using the unweighted pair group method with arithmetic mean (UPGMA) clustering algorithm. Node labels identify different clusters (1–7) and the number inside the wedge indicates the number of soil samples in a cluster. Pie charts indicate the percentage of samples that are located above the kimberlite (blue segments) and percentage of samples that are located above background (beige segments) for each cluster. **d** Response ratios of geochemical pathfinder elements compared to suites of indicator species derived from microbial community fingerprinting at DO-18. Response ratios are expressed in percent (%) calculated by the average "on deposit" (soils above kimberlite) over the average "off deposit" (soils above background) relative to an equivalent ratio of 1. **e** A microbial anomaly map shows the normalized sum of incubation-predicted indicator species' spatial distribution at DO-18. **f** A microbial anomaly map shows the normalized sum of incubation-predicted indicator species and DO-18 predicted indicator species' spatial distribution at DO-18. Indicator species in (**e**, **f**) are based on a LEfSe indicator species analysis. **g** A geochemical anomaly map at DO-18 shows the normalized sum of pathfinder elements Cr, Mg, Nb, and Ni. Results are derived from 4-acid digests and ICP-MS determination of elements from b-horizon soils. In each map (**e–g**), data (multi-colored bubbles) overlies a surficial materials map derived from field observations. Individual indicator species (**e**, **f**) or pathfinder elements (**g**) were normalized to the mean prior to summation and anomaly intervals are based on probability plots, where the "*" represents soil samples that correspond spatially to "on deposit" (above kimberlite) (**e–g**).

Kelvin are broadly similar to those at DO-18 at the phylum level (Figs. 2a and 3a). In terms of the most abundant taxa at the species level, 14 of the top 20 (70%) most abundant taxa are identical, implying that soils from the two sites are similar in the more abundant taxa but diverge in the rarer members. As with DO-18, phylum-level distributions were relatively homogenous across the sampling grid, but variability was observed at the species level, and this variability could be geographically linked to the surface expression of the buried kimberlite (Fig. 3a–c, e, f).

The application of our combined suite of 78 indicator species (summed relative abundance 7 ± 4%, Supplementary Data 7) developed through both incubation experiments and statistical analyses at DO-18 led to anomaly delineation that precisely resolved the geographic location of the underlying kimberlite mineralization at Kelvin (Fig. 3e). Again, for comparison, we also analyzed a suite of geochemical indicators (Nb, Cr, Ni, Mg), which yielded erratic anomalies that are discordant with the surface expression of the underlying kimberlite (Fig. 3d, g).

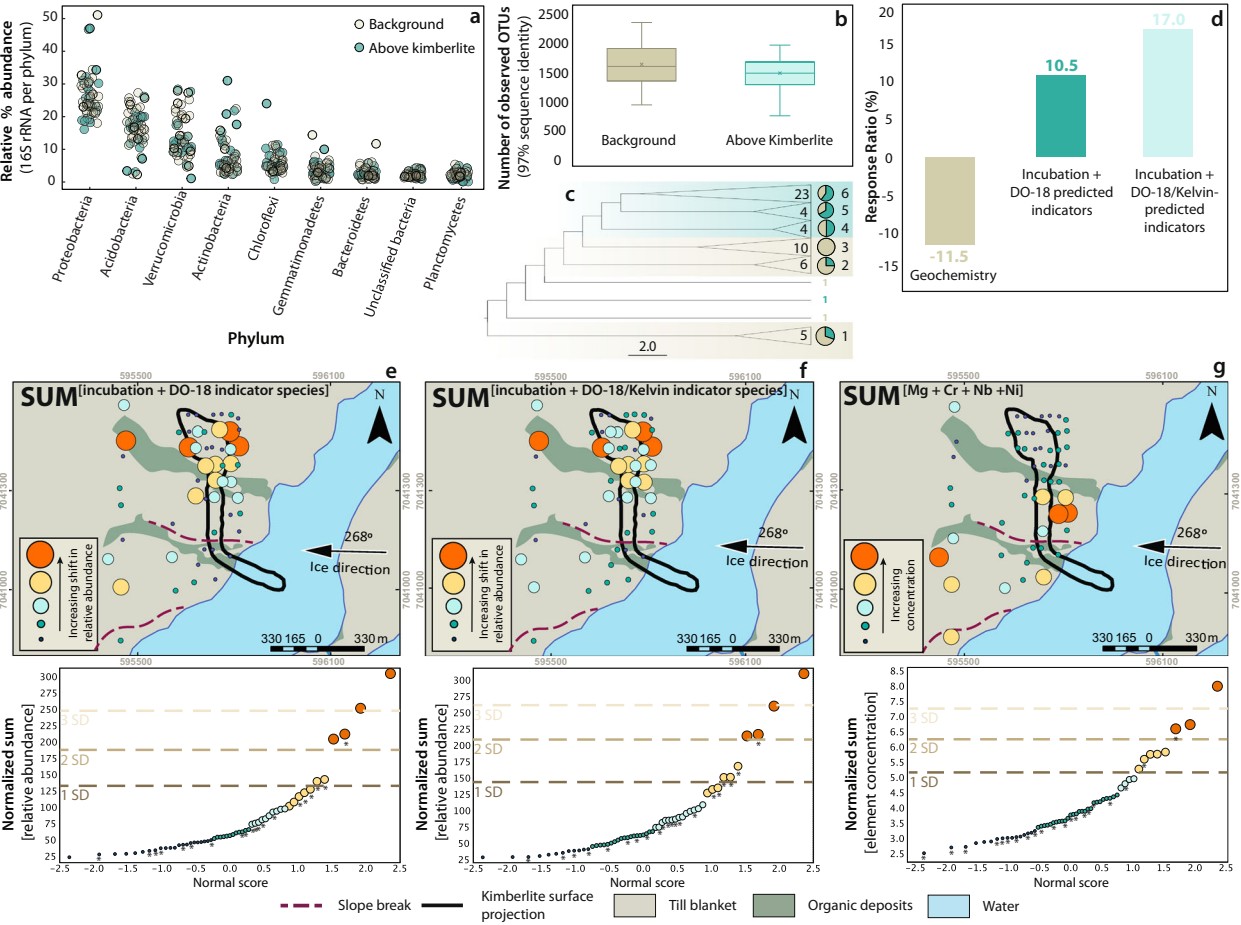

**Fig. 3 Soil microbial community composition, diversity, and indicator species for the Kelvin kimberlite. a** Distribution of 16 S rRNA gene reads per phylum for each sample at the Kelvin kimberlite. The number of reads per phylum is calculated as a percentage of the total reads for each sample. **b** Box plots show the number of observed operational taxonomic units (OTUs; 97% sequence similarity) across background soils and soils above the Kelvin kimberlite (from data that has been rarefied to 16,365 sequences per sample). Median values are indicated by the solid line within each box, and the box extends to upper and lower quartile values. Stars indicate the mean. Error bars represent standard deviation. **c** Hierarchical relationships amongst soil samples are based on Euclidean distance of 16S-OTU abundances. The hierarchical relationships between soil samples were obtained using the unweighted pair group method with arithmetic mean (UPGMA) clustering algorithm. Node labels identify different clusters (1–6) and the number inside the wedge indicates the number of soil samples in a cluster. Pie charts indicate the percentage of samples that are located above the kimberlite (blue segments) and percentage of samples that are located above background (beige segments) for each cluster. **d** Response ratios of geochemical pathfinder elements compared to suites of indicator species derived from microbial community fingerprinting at Kelvin. Response ratios are expressed in percent calculated by the average "on deposit" (soils above kimberlite) over the average "off deposit" (soils above background) relative to an equivalent ratio of 1. **e** A microbial anomaly map shows the normalized sum of incubation-predicted indicator species and DO-18 predicted indicator species' spatial distribution at Kelvin. **f** A microbial anomaly map shows the normalized sum of incubation-predicted indicator species, DO-18 predicted indicator species, and Kelvin-predicted indicator species' spatial distribution at Kelvin. Indicator species in (**e**, **f**) are based on a LEfSe indicator species analysis. **g** A geochemical anomaly map at Kelvin shows the normalized sum of pathfinder elements Cr, Mg, Nb, and Ni. Results are derived from 4-acid digests and ICP-MS determination of elements from b-horizon soils. In each map (**e**–**g**), data (multi-colored bubbles) overlies a surficial materials map derived from field observations. Individual indicator species (**e**, **f**) or pathfinder elements (**g**) were normalized to the mean prior to summation and anomaly intervals are based on probability plots, where the "*" represents samples that correspond spatially to "on deposit" (above kimberlite) (**e**–**g**).

Therefore, like at DO-18, DNA sequence-based microbial community profiling at Kelvin more effectively resolves buried mineralization than geochemical analyses. Application to Kelvin, furthermore, demonstrates that microbial community indicators developed at one deposit can be applied to the discovery of other deposits with similar microbial community compositions, at least in the same soil terrains or ecoregions. It further implies that the development of databases of indicator species can improve the use of microbial community profiling as an exploration tool. To illustrate this point, we conducted an indicator species analysis for Kelvin, as we did for DO-18 above, and this yields an additional 8 indicator species, of these one is common to DO-18 (Fig. 3f). Our analyses at Kelvin thus demonstrate capacity for

blind discovery of kimberlite mineralization buried under 10 s of meters of overburden using DNA sequencing-based microbial community analyses. These analyses can be used as a means for effectively defining drill targets in deposit to regional-scale phases of mineral exploration.

## Conclusions

Our demonstration that DNA sequences from soil microbial communities effectively resolve buried mineralization illustrates how modern sequencing technology can be leveraged in resource exploration. This finding, foremost, shows that DNA sequencing of soil microbial communities can be used in the discovery of new mineral deposits, which, by analogy to the development and

widespread application of geochemical tools to mineral exploration in the 1970's, may catalyze new deposit discovery in the decades to come. This has potential to promote the discovery of new kimberlite bodies, which could be utilized as source rocks for atmospheric carbon sequestration as well as for their stores of gem and industrial-grade diamonds. DNA sequencing of soil microbial communities also has potential application across a broad array of metallic deposits, like porphyry-type copper deposits, for which the greatest mineral potential exists in terrains with thick cover such as northern Chile and British Columbia, Canada. This should be tested through further research. More broadly, that microbial community compositions can provide better-resolved indicators of subsurface geology than geochemical analyses underscores the idea that microorganisms are acutely sensitive to their surroundings and respond to parameters that may themselves be only poorly resolved through use of even our most sophisticated existing analytical tools. Nevertheless, future research should be directed at deciphering the mechanistic link between surface soil microbial communities and underlying geology, the capacity of the approach to differentiate between kimberlite lithologies, and suitability across a wide range of soil and mineral resource types. Use of DNA sequences from microbial communities as vectors towards buried ore mineralization represents a powerful example of how such microbial information may become essential for meeting future human resource needs.

## Methods

**Geologic setting**. The DO-18 kimberlite is a Group I kimberlite that is part of the Tli Kwi Cho kimberlite complex in the Lac de Gras kimberlite field of the Archaean Slave Craton in northern Canada (Supplementary Fig. 4). It is a classic carrot-shaped kimberlite primarily composed of pyroclastic kimberlite (PK), with less dominant phases of re-sedimented volcaniclastic kimberlite (RVK), that intruded into undifferentiated Archaean granitoids[59-61]. Sedimentary mudstones and terrestrial palynomorphs that infill the kimberlite constrain the age emplacement to between 75 Ma and 45 Ma (Late Cretaceous to Eocene) at the northernmost stand of the Western Interior Seaway[61]. DO-18 is concealed by 5–20 m of glacial till (Supplementary Fig. 3a) that was deposited during the most recent late Wisconsinan glaciation by westward flow (290–295°)[62,63]. The DO-18 kimberlite has an expression of 4 ha at the till-bedrock interface[61].

The Kelvin kimberlite is also hosted within the Slave Craton of northern Canada (Supplementary Fig. S3), as one of four gently dipping, irregular L-shaped pipes that make up the Kelvin–Faraday Corridor (KFC) cluster[64,65]. It is composed of eight separate kimberlite phases of early Cambrian age, that are dominantly Kimberley-type pyroclastic kimberlite (KPK) with lesser hypabyssal kimberlite (HK), hosted within metaturbidites of the Yellowknife supergroup[64,66]. The Kelvin body is concealed under 150 m of bedrock at its northernmost extent, with the only "outcropping" rock located beneath Kelvin Lake (0.08 ha)[64,65]. The Kelvin kimberlite is further buried beneath a relatively thin (4 m) till blanket (Supplementary Fig. 3b) that was glacially deposited in the late Wisconsinan, with the most recent direction of glacial flow at 268°[67].

**Geochemical profiles**. Traditional surface-based geochemical techniques for kimberlite exploration have historically been employed by identifying geochemical signatures down-ice from kimberlites through various near-total acid soil digestions. However, the geochemical gradients of pathfinder elements linked to kimberlites can often be too subtle for reliable detection. A suite of indicator and pathfinder elements from these analyses

are typically utilized to find buried targets including Ni, Cr, Ba, Co, Sr, Rb, Nb, Mg, Ta, Ca, Fe, K, Ti, and rare-earth element (REE) concentrations, but their application depends on knowing the wide range of kimberlite host rock compositions. At DO-18 and Kelvin, anomalous concentrations of Cr, Ni, Nb, and Mg were found to be best spatially associated with the down-ice distribution of kimberlite materials in till (Figs. 2g and 3g and Supplementary Data 8a, b). A sum of Cr, Ni, Nb, Mg concentrations (Figs. 2g and 3g and Supplementary Data 8a, b) to a non-parametric normalized scale enhances the signal giving increased confidence in the likelihood of a subsurface kimberlite. The primary elements at DO-18 and Kelvin are controlled by the weathering of dominant minerals during clastic dispersion including olivine ($(Mg,Fe)_2SiO_4$), chromite ($FeCr_2O_4$), pyrope ($Mg_3Al_2Si_3O_{12}$) and picroilmenite ($FeTiO_3$) for Cr; olivine; picroilmenite and pyrope for Mg; picroilmenite for Nb; and olivine, picroilmenite and chromite for Ni. At both Kelvin and DO-18, geochemical anomalies in Cr, Ni, Nb, Mg in till generated by mechanical glacial dispersion are concentrated in the down-ice direction (Figs. 2g and 3g and Supplementary Data 8a, b) and to lesser extent above the kimberlite. This technique allows for vectoring towards a potential kimberlite via mineral and element trains in till, but does not typically delineate the target directly.

**Field sampling and QA/QC**. Sampling grids were established over known diamondiferous kimberlite pipes in accordance with standard practice in the mineral resource exploration industry (Supplementary Fig. 3a, b). The grid was designed such that it captured up-ice background materials, down-ice background materials and materials that directly overly the surface expression of the pipes. Soils for microbial community analysis at DO-18 and Kelvin were sampled with sanitized equipment without field screening, to preserve the microbial community as much as possible. Descriptions were documented for in situ physico-chemical variables at each sample site for every observed soil horizon in the profile. Soils at the field sites are derived from the breakdown of till by surface-weathering processes in situ, so the soils are considered residual weathering products of the till blanket. The B-horizon soils were targeted for microbial soil samples, although multiple horizons (including O, Ah, Ae, and C) were taken, where possible, for future analyses. Soil samples were frozen at −20 °C upon return to the laboratory at The University of British Columbia (UBC) after 1–2 weeks in field storage and transit, prior to DNA extraction. Sub-samples of the soils used in microbial community profiling were also collected for geochemical analysis. Field measurements consisted of slurry tests for pH and oxidation-reduction potential (ORP) after field sieving to below 180 μm. Samples (~1 kg) were sent to ALS Minerals Laboratories Ltd. (North Vancouver, BC) for multi-acid digestion and subsequent elemental concentration analysis via ICP-MS. Field duplicates, CRMs (certified reference materials), and blanks were inserted into the analytical stream every 15 samples (Supplementary Data 8a, b).

**Kimberlite amendment soil incubation experiments**. A bulk soil sample from the Kelvin area with background-level metal concentrations was collected from the upper B-horizon under aseptic conditions. The soil was packed into a sealed Poly Ore sample bag and stored at ambient temperature in the field. The soil was digested using a multi-acid near-total digestion and the digestate analyzed by inductively coupled plasma–mass spectrometry (ICP-MS) to determine that the soil contained 15 ppm Cr, 0.24% Mg, 7 ppm Ni, and 2 ppm Nb. The bulk soil was not dried prior to the start of the experiment. We amended tundra-derived soils with pulverized kimberlite (80% passing 10 mesh (2 mm)) by

mixing the kimberlite into the soil with a scupula. Kimberlite size was based on our field observations that the glacial till soil matrix is dominated by sand sized particles (0.06–2 mm). Control soils were similarly mixed without the addition of kimberlite. Soil was dispensed aseptically into sterile containers for each treatment with amendment concentrations chosen to represent concentrations of pathfinder elements that are routinely detected in geochemical surveys over buried mineral deposits (5% dilution). Soil was sampled at T = 0, T = 1 (15 days), T = 2 (55 days), and T = 3 (85 days).

**DNA extraction and QA/QC.** DNA was extracted using a DNeasy PowerSoil Kit (Qiagen). The resulting DNA was stored at −20 °C. DNA was quantified using the PicoGreen® Assay (Invitrogen) for dsDNA and measured on a TECAN™ M200 (excitation at 480 nm and emission at 520 nm). The purity and quality of the extracted DNA was assessed based on the ratio of absorbance at 260 nm to absorbance at 280 nm, which were measured using a NanoDrop® ND-1000 spectrophotometer (Thermo Scientific).

**SSU rRNA gene amplification and DNA amplicon sequencing.** Bacterial and archaeal SSU rRNA gene fragments (V4 region) were amplified from the extracted genomic DNA using primers 515 F and 806 R[44,68]. Sample preparation for amplicon sequencing was performed as described in refs. [44] and [68]. In brief, the aforementioned SSU rRNA gene-targeting primers, complete with Illumina adapter, an 8-nt index sequence, a 10-nt pad sequence, a 2-nt linker, and the gene-specific primer were used in equimolar concentrations of 0.2 μm together with dNTPs, PCR buffer, MgCl₂, 2U/ul ThermoFisher Phusion Hot Start II DNA polymerase, and PCR-certified water to a final volume of 25 L. PCR amplification was performed with an initial denaturing step of 95 C for 2 min, followed by 30 cycles of denaturation (95 °C for 30 s), annealing (55 °C for 30 s), and elongation (72 °C for 1 min), with a final elongation step at 72 °C for 10 min. Equimolar concentrations of prepared amplicon-bearing solutions were pooled into a single library by using the Invitrogen SequalPrep kit. The amplicon library was analyzed on an Agilent Bioanalyser using the High Sensitivity dsDNA assay to determine approximate library fragment size, and to verify library integrity. Pooled library DNA concentration was determined using the KAPA Library Quantification Kit for Illumina. Library pools were diluted to 4 nM DNA, which was denatured into single strands using fresh 0.2 N NaOH, as recommended by Illumina. The final library was loaded at a concentration of 8 pM DNA, with an additional PhiX spike-in of 5–20%. Sequencing was conducted with MiSeq at the UBC sequencing center.

**Bioinformatics.** DNA sequences were processed using the Mothur amplicon sequence analysis pipeline[69]. Sequences were removed from the analysis if they contained ambiguous characters, had homopolymers longer than 8 bp, or did not align to a reference alignment of the sequencing region. Unique sequences and their frequencies in each sample were identified and then a pre-clustering algorithm was used to further de-noise sequences within each sample[70]. The unique sequences were aligned against the SILVA reference alignment (available online at https://mothur.org/wiki/silva_reference_files/). Sequences were chimera checked using vsearch[71,72] and reads were then clustered into 97% OTUs using OptiClust[73]. OTUs were classified using SILVA reference taxonomy database (release 132, available online at https://mothur.org/wiki/silva_reference_files/). OTUs that had less than 2 reads were filtered from analysis. For alpha and beta diversity measures, all samples were subsampled to the lowest coverage depth (16365) and calculated in Mothur[69]. Chao1 was

calculated from filtered data and thus effectively represents a rarefaction of the observed OTUs. Sequences were deposited into the Sequence read archive (SRA) under accession number PRJNA698256.

**Anomaly identification and mapping.** Indicator species analyses (LEfSe) were performed based on algorithms defined by[74] where indicator species (OTUs) are considered significant if the LDA score >2. Sample groups for the kimberlite amendment incubation experiment are based on unamended "control soils" and amended "kimberlite-bearing soils". Indicators for the amendment were paired to known metabolic functions by using the FAPROTAX tool[55]. Sample groups were set for field analyses based on their origin from "background soil" or "soils above kimberlite". These groups are defined based on underlying geology whereby "background soils" come from above the metaturbidite (Kelvin) or granodiorite (DO-18) host rock, and "soils above kimberlite" come from above the surface projection of the kimberlites as defined by drilling.

Incubation-derived LEfSe indicator species showing an enrichment in the kimberlite-amended soil samples were curated to plot at DO-18. Indicator species with >1 average reads per sample in the incubation experiment and positive response ratios at the DO-18 field site were included. Response ratios for indicator species were calculated as the ratio between the average relative abundance in "soils above kimberlite" and the average relative abundance of "background soils". LEfSe indicator species predicted from the DO-18 and Kelvin field sites were not curated further, thus each indicator species output was included in the generation of the anomaly maps.

Map data plots were created using relative abundances of indicator species from 16 S rRNA gene sequencing and pathfinder element concentrations. Individual indicator species and pathfinder elements were normalized to the mean prior to summation. Response ratio bar plots of the normalized sums of indicator species and pathfinder elements are expressed by the following equation: $((average(\frac{on\ deposit}{off\ deposit})) - 1) * 100$. Anomaly identification through probability plots was done in the Reflex/Imdex ioGAS software (version 8.0), and mapping of anomalies and surficial geology was performed in the ESRI ArcGIS software. To determine if predictive indicators could be generated by chance, we randomized the sample group sets for field analyses based on their origin from "background soil" or "soils above kimberlite". Response ratios at the Kelvin field site were calculated based on a set of ten randomly generated Lefse results from DO-18 (Supplementary Data 9). Seven of ten of these response ratios were negative showing no spatial correlation between the bacterial anomaly and the surface projection of the Kelvin kimberlite. This shows that it is unlikely that our collection of indicator species, which display positive surface anomalies with respect to subsurface kimberlites, could be randomly generated.

**Reporting summary.** Further information on research design is available in the Nature Portfolio Reporting Summary linked to this article.

## Data availability

Sequences were deposited into the Sequence read archive (SRA) under accession number PRJNA698256. Supplementary data has been deposited into figshare, available at https://doi.org/10.6084/m9.figshare.23960790.

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

## Acknowledgements
Support for this work came from NSERC/AcmeLabs/Bureau Veritas Minerals Industrial Research Chair in Exploration Geochemistry held by P.A.W. and a Tier II Canada Research Chair in Geomicrobiology held by S.A.C. The research was further supported by an NSERC Discovery grant (0487) held by S.A.C. The Government of the Northwest Territories (GNWT) provided in-kind financial support, and Barrett Elliott (GNWT) facilitated field work. De Beers (formerly Peregrine Diamonds Ltd.) and Mountain Province Diamonds (formerly Kennady Diamonds) provided access to field sites and permission to collect soil samples from DO-18 and Kelvin kimberlites, respectively.

## Author contributions
R.L.S. conducted microbial community analyses, analyzed and interpreted data, supervised research, and wrote the manuscript with input from B.P.I.P., C.J.R.H., and S.A.C. B.P.I.P. collected samples, interpreted results from soil geochemistry, analyzed the surface expression of microbial anomalies, analyzed and interpreted the data, and supported the writing and editing process. A.P.W. collected samples, mapped the surface environment, and interpreted results from soil geochemistry. E.M.C. collected samples, mapped the surface environment, and interpreted results from soil geochemistry. C.J.R.H. supervised research and supported the writing and editing process. P.A.W. conceived and designed the research, analyzed and interpreted the data, supervised research, and supported the writing and editing process. S.A.C. conceived and designed the research, analyzed and interpreted the data, supervised research, and supported the writing and editing process.

## Competing interests
R.L.S., C.J.R.H., and S.A.C. are members of a commercial entity that offers sequencing services to the mining exploration industry and others and thus declare the existence of a financial competing interest. R.L.S., C.J.R.H., and S.A.C.'s commercial affiliations do not alter adherence to Nature Portfolio journals' policies on sharing data and materials. The authors declare no other competing interests.
