## [Peer Review File · Communications Earth & Environment]

5th Apr 23

Dear Dr Crowe,

Your manuscript titled "DNA sequencing, microbial sensing, and the discovery of buried mineral resources" has now been seen by 3 reviewers, whose comments are appended below. You will see that they find your work of some potential interest. However, they have raised quite substantial concerns that must be addressed. In light of these comments, we cannot accept the manuscript for publication, but would be interested in considering a revised version that fully addresses these serious concerns.

Additionally, I would suggest addressing the following points:

- * Present a compelling case that microbial community DNA is an effective means to identify kimberlite deposits in the subsurface
- * Adapt or justify your experimental approach and statistical analyses to ensure that they are robust and representative of real-world conditions
- * Tone down the rhetoric on the importance of kimberlite mining for sustaining the human population and ensure that your title, abstract and introduction are focused on the specific results and implications of your study

We hope you will find the reviewers' comments useful as you decide how to proceed. Should additional work allow you to address these criticisms, we would be happy to look at a substantially revised manuscript. If you choose to take up this option, please either highlight all changes in the manuscript text file, or provide a list of the changes to the manuscript with your responses to the reviewers.

If the revision process takes significantly longer than three months, we will be happy to reconsider your paper at a later date, as long as nothing similar has been accepted for publication at Communications Earth & Environment or published elsewhere in the meantime.

We understand that due to the current global situation, the time required for revision may be longer than usual. We would appreciate it if you could keep us informed about an estimated timescale for resubmission, to facilitate our planning. Of course, if you are unable to estimate, we are happy to accommodate necessary extensions nevertheless.

Please use the following link to submit your revised manuscript, point-by-point response to the reviewers' comments with a list of your changes to the manuscript text (which should be in a

separate document to any cover letter) and any completed checklist:

[link redacted]

Please do not hesitate to contact me if you have any questions or would like to discuss the required revisions further. Thank you for the opportunity to review your work.

Best regards,

Sadia Ilyas
External Editor
Communications Earth & Environment

Joe Aslin
Senior Editor
Communications Earth & Environment

EDITORIAL POLICIES AND FORMAT

If you decide to resubmit your paper, please ensure that your manuscript complies with our editorial policies and complete and upload the checklist below as a Related Manuscript file type with the revised article:

Editorial Policy Policy requirements (Download the link to your computer as a PDF.)

For your information, you can find some guidance regarding format requirements summarized on the following checklist:(<https://www.nature.com/documents/commsj-phys-style-formatting-checklist-article.pdf>) and formatting guide (<https://www.nature.com/documents/commsj-phys-style-formatting-guide-accept.pdf>).

REVIEWER COMMENTS:

Reviewer #1 (Remarks to the Author):

Overview:

This manuscript by Simister and colleagues describes novel research testing the potential application of microbial DNA amplicon sequencing to mineral resource exploration. More specifically, their study first examines responses in the microbial community of an arctic tundra soil to kimberlite ore

additions under laboratory conditions, and successfully apply these findings to delineate buried kimberlite ore bodies in field settings. The authors also compare microbial results to geochemical results obtained using traditional exploration approaches and considered both informed and blind applications of this DNA-based exploration strategy. Overall, the results demonstrate that microbial communities respond to the presence of ore minerals under laboratory conditions, and similar responses can be detected under field conditions. These results clearly demonstrate potential for application of microbial DNA amplicon sequencing as a mineral exploration tool.

The manuscript is very well written and was a pleasure to read. The introduction and methods sections include sufficient detail for readers to understand the research motivation and approach. The results section clearly conveys the principal experimental approaches and analytical methods, both of which are appropriate for this study. Data interpretation is supported by both field and laboratory results, and the discussion clearly relates these results to the current state of knowledge. This manuscript presents compelling results that should be of broad interest to Nature Communications Earth & Environment readers and beyond. I have provided some comments and questions below for consideration if revisions are requested.

Comments/Questions:

Abstract: I strongly agree with the first sentence, but one could argue that exploration, discovery, and mining of kimberlite ores is absolutely not required to “sustain human population growth and technological advancements that will enable global decarbonization”. Consequently, it is important to reconcile method development applied to kimberlite ores with broad application to other commodities.

Lines 62-64: Although I am not disputing the content of these references, an argument can be made that established mineral resources and reserves, which often grow during production, are sufficient to meet societal demand for the foreseeable future. Instead, one could argue that projected supply issues stem from challenges securing funding and permits to increase production capacity. This is not to say that new discoveries are not important, but this statement implies a more dire situation than currently exists.

Lines 67-70: I was surprised that the authors did not relate this research to the (bio)geobattery model, which is not a new concept in mineral exploration. Do the authors think their observed microbial community response is related to a (bio)geobattery system, where surface expression of subsurface microbial redox processes can be detected in microbial communities?

Lines 88-91: This sentence implies that mining kimberlites to produce ultramafic mine wastes to support carbon capture is an important aspect of diamond mining. Perhaps I am missing something, but I have yet to see evidence that mineral carbonation in ultramafic mine wastes could completely offset carbon emissions associated with that mine. It is difficult to see the value in diamond mining to enhance carbon sequestration, particularly when other environmental impacts of mining are fully considered.

Line 105: I am curious why 5% (w/w) kimberlite addition was selected as it seems high from an “indicator mineral” perspective? Was there any analysis of tundra soils performed to validate this amendment rate? Were any experiments performed at lower and higher amendment rates to identify a response threshold?

Line 118: Specifically at the 5% (w/w) amendment rate.

Line 125: Relative decline?

Line 142: Relative abundance?

Lines 143-145: Any thoughts why these specific taxa responded? Are these responses likely specific to arctic tundra soils or kimberlite ore?

Line 233: "Similar but different" is a very ambiguous statement. Potential to elaborate?

Line 243: Geochemical surveys are rarely performed independent of other approaches (e.g., geophysics, drilling).

Reviewer #2 (Remarks to the Author):

The present manuscript underpins the potential use of DNA sequencing of microbes to be applied in the exploration of mineral resources that remained hidden below the ground. This is a new technique to employ for identifying the new reserves of minerals, however, in the recent past have been applied by some researchers. This manuscript would be further helpful to understand such application in more detail, albeit, it requires a significant revision in terms of the technicality and the presentation of the manuscript. My detailed comments are given below.

Comments:

1. Title: The present title is too general; hence, it should be modified so that one could get the clear idea on the study focused in this manuscript.

2. Abstract: Although minerals are an essential part of our day-to-day lives in certain ways, the connection of kimberlite ore with the sustainability of the human population is somehow awkward. This reviewer would like to suggest deleting 2 sentences of the Abstract and re-construct the entire section by adding more relevant data and findings from this study by keeping in mind that an abstract is always a standalone part of the manuscript to provide a glimpse of the entire workout.

3. Introduction: After going through this section, the reviewer is somehow not happy in the way the introduction is narrated. I would like to suggest connecting this piece of study as a part of bio-mining. Traditionally, it has been used for the bio-oxidation and bio-accumulation of a targeted mineral and applied at commercial scale of chalcopyrite, gold-bearing arsenopyrite, and to some oxide mineral deposits as well. But the use of microorganisms can further be extended to identify the mineral reserves hidden at the bottom of the ground surface by knowing some important characteristics of the microorganism's culture at the surroundings, wherein, DNA analysis can provide ample information to lead in such direction. As I said that although the manuscript has potential, it needs to give some angle to significantly differentiate this study with another and then to present a solid hypothesis on the work done.

4. Furthermore, the authors need to clarify here that why they applied microbial communities can have the potential role over other bioinformatic techniques. The authors should present some

comparative analysis with other techniques.

5. Line 103: what is the basis of the considered invariable of using -10 mesh soils and 5wt.% of kimberlite for the soil amendment? What if we use another mesh size and/or different wt% of kimberlite for the soil amendment? Is this a limitation of the study? The authors need to clarify these all and a solid explanation on their considerations. Whether they have any previous study/data to refer to justify their preliminary considerations?

6. Line 138-160: There is no explanation given behind the change in taxonomy. Moreover, it needs to know whether this variance is exclusive with the kimberlite that is used in this study or occurs with any other sample as well.

7. Line 259-261: The reviewer would like to see more evidence for saying such a statement. I strongly feel that the authors should be fixed on their focused study. Why suddenly the carbon sequestration etc. are discussed. Whether this study is directly applicable to Cu-porphyry ores? If not then better to not name any specific mineral. It's commonly understood that the idea can be examined with any other mineral deposits at any other area.

8. This reviewer has some grievances on geochemical analysis and would like to present it in a more effective manner when comparing with the bioinformatics and conclusions are made on that.

9. Can a SWOT analysis can be presented in tabular form to describe the better suitability of the applied technique?

Reviewer #3 (Remarks to the Author):

The overarching claim of the paper is that it is possible to more accurately predict the subsurface presence of kimberlite through bacterial community composition of the surface soil than by using chemical indicators. Application of such microbial tracking for kimberlite and developing a library of indicator organisms that enrich in the presence of kimberlite is new. However, the approach overall has been applied in oil exploration and EOR. Some relevant references are <https://doi.org/10.1016/j.earscirev.2021.103563>; and <https://doi.org/10.3389/fmicb.2019.02996>. At the same time, the application for predicting the location of kimberlite is novel.

I think that this paper would be of interest to scientists and also companies interested in diamond exploration. It is a good study with relevance, however I am not confident about use of the bacteria grown in batch reactors with kimberlite to populate the list of kimberlite-indicating organisms: The objective is to find bacteria markers in surface soil when the kimberlite is in the subsurface region. In the enrichment reactors, the kimberlite and sampled soil are together and are not representative of the conditions on the ground.

On another note, I am also not impressed by more sampling sites within the zone above kimberlite compared to the control. While the statistical analysis is standard, and I have no objections to it, I am not confident in the claims of unique bacterial community structure above subsurface kimberlite deposits: the bacterial species that undergo remarkable changes are a small fraction of the whole community.

At the same time, I appreciate that the paper is well-written and easy to read.

REVIEWER COMMENTS:

Reviewer #1 (Remarks to the Author):

Overview:

This manuscript by Simister and colleagues describes novel research testing the potential application of microbial DNA amplicon sequencing to mineral resource exploration. More specifically, their study first examines responses in the microbial community of an arctic tundra soil to kimberlite ore additions under laboratory conditions, and successfully apply these findings to delineate buried kimberlite ore bodies in field settings. The authors also compare microbial results to geochemical results obtained using traditional exploration approaches and considered both informed and blind applications of this DNA-based exploration strategy. Overall, the results demonstrate that microbial communities respond to the presence of ore minerals under laboratory conditions, and similar responses can be detected under field conditions. These results clearly demonstrate potential for application of microbial DNA amplicon sequencing as a mineral exploration tool.

The manuscript is very well written and was a pleasure to read. The introduction and methods sections include sufficient detail for readers to understand the research motivation and approach. The results section clearly conveys the principal experimental approaches and analytical methods, both of which are appropriate for this study. Data interpretation is supported by both field and laboratory results, and the discussion clearly relates these results to the current state of knowledge. This manuscript presents compelling results that should be of broad interest to Nature Communications Earth & Environment readers and beyond. I have provided some comments and questions below for consideration if revisions are requested.

Thank you for the positive and thoughtful review.

Comments/Questions:

Abstract: I strongly agree with the first sentence, but one could argue that exploration, discovery, and mining of kimberlite ores is absolutely not required to “sustain human population growth and technological advancements that will enable global decarbonization”. Consequently, it is important to reconcile method development applied to kimberlite ores with broad application to other commodities.

Agreed. We didn't mean to imply that kimberlites were needed to ‘sustain human population growth and technological advancements that will enable global decarbonization’, rather that mineral resources more generally are needed. We have tried to clarify this by modifying the abstract to read:

Line 25-28 “Here we used diamondiferous kimberlite ore bodies as a test case and show that DNA amplicon sequencing of soil microbial communities resolves anomalies in microbial community composition and structure that reflect the surface expression of kimberlites buried under 10s of meters of overburden”

Lines 62-64: Although I am not disputing the content of these references, an argument can be made that established mineral resources and reserves, which often grow during production, are sufficient to meet societal demand for the foreseeable future. Instead, one could argue that projected supply issues stem from challenges securing funding and permits to increase production capacity. This is not to say that new discoveries are not important, but this statement implies a more dire situation than currently exists.

While we appreciate this, but feel the text is accurate as written. We have removed the superlative 'rapidly' to de-emphasize (Line 69).

Lines 67-70: I was surprised that the authors did not relate this research to the (bio)geobattery model, which is not a new concept in mineral exploration. Do the authors think their observed microbial community response is related to a (bio)geobattery system, where surface expression of subsurface microbial redox processes can be detected in microbial communities?

Our understanding is that the (bio)geobattery system is an interesting, though not fully tested model for the transport of mineral components from subsurface deposits to surface soils via electrochemical cells. If this model is an accurate representation of what happens, it could play a role in dictating microbial community compositions and thus microbial based mineral deposit anomalies at the surface. While we feel it is certainly something worth exploring, our sense is that there is insufficient data to support the (bio)geobattery model as a generalizable theory at this point and thus haven't discussed it here since it is somewhat tangential to the main message of our paper.

Lines 88-91: This sentence implies that mining kimberlites to produce ultramafic mine wastes to support carbon capture is an important aspect of diamond mining. Perhaps I am missing something, but I have yet to see evidence that mineral carbonation in ultramafic mine wastes could completely offset carbon emissions associated with that mine. It is difficult to see the value in diamond mining to enhance carbon sequestration, particularly when other environmental impacts of mining are fully considered.

This is a fair point, but it could be possible to mine kimberlites for their carbon capture potential exclusively and independent of diamond extraction. If such mining were powered entirely by renewables, not linked to carbon emissions, then the mineral carbon capture would more than offset the emissions of mining itself. Indeed, such approaches with ultramafic rocks, more generally, are the subject of much discussion in the carbon capture community. We believe this is a useful statement to make, however, we leave it at the editor's discretion and could remove it if requested.

Line 105: I am curious why 5% (w/w) kimberlite addition was selected as it seems high from an "indicator mineral" perspective? Was there any analysis of tundra soils performed to validate this amendment rate? Were any experiments performed at lower and higher amendment rates to identify a response threshold?

The 5% (w/w) addition is justified based on the chemical pathfinder concentrations considered anomalous in exploration. This was noted and explained in the original manuscript lines 122-123 and 371-374. We have not conducted experiments with lower or higher amendments to identify thresholds. This is something that we are currently pursuing.

Line 118: Specifically at the 5% (w/w) amendment rate.

We have added this to the text:

Line 117-120 "We found that over a period of 85 days, the microbial community composition and structure in soils amended with 5% w/w kimberlite, diverged from...."

Line 125: Relative decline?

We have clarified that this is relative to the baseline line.

Line 32-134 “This decline in species richness is also supported by a decline in the number of observed OTUs relative to baseline, which decrease by.....”

Line 142: Relative abundance?

We have clarified that this is relative abundance.

Line 148-150 “Of these, 65 species (17%) increased in relative abundance over the 85-day incubation period,....”

Lines 143-145: Any thoughts why these specific taxa responded? Are these responses likely specific to arctic tundra soils or kimberlite ore?

Many/most taxa that responded significantly (i.e. the indicator species) are only broadly related to taxa represented in culture collections. Response of specific taxa to amendment is likely linked to differences in physiology/function. It is notoriously difficult, however, to infer function from taxonomy, particularly as the phylogenetic distance between taxa and lab cultures increases. We had originally chosen not to attempt to infer functional relationships from our taxonomic data (and this is generally against best practices), but recognizing the limits of the available data we have now conducted a high-level analysis of taxonomic/functional relationships using existing tools. The results are tenuous, but we have qualified our interpretations with what we feel are the appropriate cautionary statements.

In an attempt to link our indicator species to their underlying metabolic potential we compared the indicator species to the FAPROTAX (Louca et al 2016) database, which maps prokaryotic clades (e.g. genera or species) to established metabolic or other ecologically relevant functions, using current literature on cultured strains. This analysis linked 16.8% of incubation indicators, organisms with known metabolic functions. These functions are common to soil microbial communities and included aerobic methanol oxidation, methylotrophy, oxidation of sulfur compounds ammonia oxidation, nitrification, nitrate respiration and reduction, nitrogen respiration, cellulolysis, chemoheterotrophy and others. While it is possible that these specific functions are somehow altered through kimberlite amendment there are likely a myriad of other functions not captured through the FAPROTAX analysis that could have easily have been influenced by exposure to the kimberlite. Instead, we feel that the FAPROTAX analysis speaks to the general potential for exposure to kimberlite to alter community composition and metabolism.

We have added the following text to the manuscript:

Line: 153-157 “Comparisons of the indicator species to a database of microbial functions [55] imply that the indicator species are associated with a wide range of metabolic potentials that are common and widely distributed in soil microbial communities (Table 3a, b). It is important to point out, however, that inferences of function from taxonomy are prone to error that arises from strong differences in metabolic potential across closely related taxa”

Line 233: “Similar but different” is a very ambiguous statement. Potential to elaborate?

We have changed the sentence to read:

Line: 243-245 “In terms of the most abundant taxa at the species level, 14 of the top 20 (70%) of the most abundant taxa are identical, implying that soils from the two soils are similar in the more abundant taxa but diverge in the rarer members”

Line 243: Geochemical surveys are rarely performed independent of other approaches (e.g., geophysics, drilling).

This is true and we envision that microbiological surveys would be performed in concert with other approaches and rather than providing a standalone tool, act as an additional layer of information when exploring for concealed mineral deposits.

Reviewer #2 (Remarks to the Author):

The present manuscript underpins the potential use of DNA sequencing of microbes to be applied in the exploration of mineral resources that remained hidden below the ground. This is a new technique to employ for identifying the new reserves of minerals, however, in the recent past have been applied by some researchers. This manuscript would be further helpful to understand such application in more detail, albeit, it requires a significant revision in terms of the technicality and the presentation of the manuscript. My detailed comments are given below.

Thank you for the thoughtful review and helpful suggestions. To our knowledge this is the first application of DNA-based microbial community profiling to the exploration for mineral resources. We haven't made such a claim in the manuscript as we feel that claims of priority distract from the scientific content and merit of a manuscript.

Comments:

1. Title: The present title is too general; hence, it should be modified so that one could get the clear idea on the study focused in this manuscript.

Given that this is the first demonstration of microbial community fingerprinting for the exploration of concealed mineral resources, and the application to kimberlites was meant as a case study (this is now clarified in the manuscript), we feel that the general title is appropriate. We have nevertheless changed “sensor” to “indicators”, to tether the title more directly to the manuscript results. Here, we will defer to the editor in terms of the suitability of the title and are happy to revise if requested.

2. Abstract: Although minerals are an essential part of our day-to-day lives in certain ways, the connection of kimberlite ore with the sustainability of the human population is somehow awkward. This reviewer would like to suggest deleting 2 sentences of the Abstract and re-construct the entire section by adding more relevant data and findings from this study by keeping in mind that an abstract is always a standalone part of the manuscript to provide a glimpse of the entire workout.

The first two sentences are meant as a generalization that reflects the extensibility of our research to mineral deposits beyond kimberlites and so we have elected to keep those two sentences, but have also elected to make it more clear in the abstract that the study of kimberlites is meant as a test case (line 26). The abstract is 10 lines long and only the first two sentences are background information, the rest reports

the findings of our work.

3. Introduction: After going through this section, the reviewer is somehow not happy in the way the introduction is narrated. I would like to suggest connecting this piece of study as a part of bio-mining. Traditionally, it has been used for the bio-oxidation and bio-accumulation of a targeted mineral and applied at commercial scale of chalcopyrite, gold-bearing arsenopyrite, and to some oxide mineral deposits as well. But the use of microorganisms can further be extended to identify the mineral reserves hidden at the bottom of the ground surface by knowing some important characteristics of the microorganism's culture at the surroundings, wherein, DNA analysis can provide ample information to lead in such direction. As I said that although the manuscript has potential, it needs to give some angle to significantly differentiate this study with another and then to present a solid hypothesis on the work done.

We appreciate the reviewer's perspective on the narrative; however, it is in direct contrast to the responses from the other reviewers. We therefore have elected to retain the narrative in its original form. The reviewer, however, raises a good point about the application of microbiology in the mineral resource sector more generally, and therefore we have added text that highlights some of these other aspects.

Line 50-52 “Through their metabolism, microorganisms affect the distribution of minerals at Earth's surface, and in extreme cases, can even lead to the formation of mineral resources.

Line 61-69 “Historically, the application of microbiology in the natural resource sector has mostly been limited to mineral processing, predominantly bioleaching of sulfide ores [18]. For example, acidophilic microorganisms, such as *Acidithiobacillus ferrooxidans* [19], which are abundant in natural environments associated with pyritic ore bodies, coal deposits[20], and their associated acid mine drainages [21], have been harnessed at commercial scale to extract copper and gold from sulfide ores for decades[22]. In contrast, we have mostly overlooked the potential power of microbial communities to enable resource discovery in the natural environment and are only just beginning to harness the capacity of environmental microbial communities as environmental resource indicators”

We believe our hypothesis is clearly stated in the first sentence of the last paragraph of the introduction (Line: 90-91) ... “soil microbial communities in sub-arctic tundra respond to and thus indicate ore materials and buried mineralization”. In our opinion this hypothesis is “solid”, and we are unaware of any other studies that report use of high-throughput sequencing approaches to identify indicator species and detect microbial community anomalies related to subsurface mineralization. A Google Scholar search comes up with our own pre-print and grey literature reports from our group as the only relevant results, as well as a perspective paper that has no data (add citations to geosci-bc reports and perspective).

4. Furthermore, the authors need to clarify here that why they applied microbial communities can have the potential role over other bioinformatic techniques. The authors should present some comparative analysis with other techniques.

We used microbial communities since they are sensitive indicators of the environment and we state this in lines (52-54)... “Microbial community compositions and structures are thus sensitive reflections of their habitats [4] and analyses of microbial communities can provide a wealth of information on their surrounding environments”.

We used the MOTHUR bioinformatic pipeline as stated in the methods, as this is a tool that is well designed for the analysis of soil microbial communities and employs a comprehensive and robust set of statistical analyses. We could have elected to use the QIIME2 pipeline, but the environmental

microbiology community is divided over the relative merits of the two pipelines and decisions on which pipeline to use mostly come down to personal preference. For our own edification, we have also used QIIME2 in related studies and find, as expected, that we get comparable results. We are not sure what other bioinformatic techniques the reviewer might be referring to.

5. Line 103: what is the basis of the considered invariable of using -10 mesh soils and 5wt.% of kimberlite for the soil amendment? What if we use another mesh size and/or different wt% of kimberlite for the soil amendment? Is this a limitation of the study? The authors need to clarify these all and a solid explanation on their considerations. Whether they have any previous study/data to refer to justify their preliminary considerations?

We sieved the kimberlite with 80% of material passing 10 mesh (2 mm) to approximate the soil matrix grain size found for the glacial till in our field areas, which based on field observations, is dominated by sand sized particles (0.06-2 mm). This information has been added to the appropriate methods section.

Line 383-385 “Kimberlite size was based on our field observations that the glacial till soil matrix is dominated by sand sized particles (0.06mm-2mm). “

The 5% (w/w) addition is justified based on the chemical indicator concentrations considered anomalous in exploration. This was noted in the original manuscript “lines 385-388”. We have not conducted experiments with lower and higher amendments to identify thresholds. This is something, however, that we are currently pursuing.

6. Line 138-160: There is no explanation given behind the change in taxonomy. Moreover, it needs to know whether this variance is exclusive with the kimberlite that is used in this study or occurs with any other sample as well.

The explanation given for the change in community was due to the amendment with kimberlite. In response to reviewer #1 we have also conducted additional analyses to assess the functional potential of the indicator species, given that the responses are likely linked to physiology (lines (153-157)). These necessarily remain speculative at this point. As also pointed out, to reviewer #1 assigning function based on taxonomy is tenuous and given that the associated functions are common to soil microorganisms this analysis provides little additional information.

We conducted the experiments in triplicate with three different soil samples and thus we get the same effect on multiple samples. We have added an additional supplementary figure that shows the phylum distribution for each soil replicate (Sup Fig 1). We only explored kimberlite amendments in this study and responses to other rock types are the subject of ongoing research. As a teaser, community responses are different with amendments with different rock types.

7. Line 259-261: The reviewer would like to see more evidence for saying such a statement. I strongly feel that the authors should be fixed on their focused study. Why suddenly the carbon sequestration etc. are discussed. Whether this study is directly applicable to Cu-porphyry ores? If not then better to not name any specific mineral. It's commonly understood that the idea can be examined with any other mineral deposits at any other area.

Here, we mostly disagree as noted above and in response to the other reviewers. The kimberlite resource was selected as a test case meant to demonstrate the broader application of the approach. Clearly, the approach needs to be validated on other mineral deposit types and this is the subject of our ongoing research. Given that this current research has only been published in peer-reviewed technical reports and not primary literature, we elected not to include citations to this work in the main text.

Please see these references/links to our peer-reviewed reports:

https://cdn.geosciencebc.com/project_data/GBCReport2023-01/Sch_IulianellaPhillips_MineralsSoA2022.pdf

https://cdn.geosciencebc.com/pdf/SummaryofActivities2021/Minerals/Sch_Phillips_MineralsSOA2021.pdf

https://cdn.geosciencebc.com/project_data/GBC%20Report2020-03/GBC%20Report%202020-03%20Microbial-Community%20Fingerprints%20as%20Indicators%20for%20Buried%20Mineralization%20in%20British%20Columbia_revised_BIP.pdf

https://cdn.geosciencebc.com/pdf/SummaryofActivities2018/MM/2016-025_Schol_SoA2018_MM_SimisterAndPhillips.pdf

8. This reviewer has some grievances on geochemical analysis and would like to present it in a more effective manner when comparing with the bioinformatics and conclusions are made on that.

The geochemical data is presented in a manner consistent with how soil geochemistry is commonly presented in mineral exploration programs and in mineral exploration research papers. It is also presented in the same way as the microbial indicator species to allow for direct comparison. The tabulated data is included in the supplement (Tables 7a, b (now tables 8a, 8b), to promote reproducible research and as such, any reader can replot the data in any way they wish to view it. Given that there are no specific recommendations as to how an alternative presentation might better display the results and that the other reviewers are happy with the presentation format, we have elected to retain the original figures.

9. Can a SWOT analysis can be presented in tabular form to describe the better suitability of the applied technique?

This is an interesting suggestion and something we believe should be done in a commercial context where mineral exploration companies are deciding whether or not to adopt microbial fingerprinting in their exploration programs. We can create a SWOT analysis and we'll leave it at the editor's discretion as to whether this is suitable at this journal.

Reviewer #3 (Remarks to the Author):

The overarching claim of the paper is that it is possible to more accurately predict the subsurface presence of kimberlite through bacterial community composition of the surface soil than by using chemical indicators. Application of such microbial tracking for kimberlite and developing a library of indicator organisms that enrich in the presence of kimberlite is new. However, the approach overall has been applied in oil exploration and EOR. Some relevant references are <https://doi.org/10.1016/j.earscirev.2021.103563>; and <https://doi.org/10.3389/fmicb.2019.02996>. At the same time, the application for predicting the location of kimberlite is novel.

We have added a reference to the use of microbial indicators in oil and gas exploration:

line 57-59 “Sequence based microbial community analyses, for example, have been used to detect organic and inorganic contaminants in groundwater at the watershed-scale [8]. They have also been used as pathfinders in petroleum exploration [11].

I think that this paper would be of interest to scientists and also companies interested in diamond exploration. It is a good study with relevance; however I am not confident about use of the bacteria grown in batch reactors with kimberlite to populate the list of kimberlite-indicating organisms:

We too were somewhat surprised that the lab-based indicator species performed effectively in the survey experiments. However, it makes sense given that many of these indicators are the same as the indicators that come out of the survey experiments, as discussed in the text.

The objective is to find bacteria markers in surface soil when the kimberlite is in the subsurface region. In the enrichment reactors, the kimberlite and sampled soil are together and are not representative of the conditions on the ground.

Further to our response above, the surface soils in the field are in fact admixtures of kimberlite material and glacially derived soils, though the kimberlite fraction is often low and potentially not directly resolvable through standard geochemical and mineralogical analyses.

On another note, I am also not impressed by more sampling sites within the zone above kimberlite compared to the control.

Both field surveys have more soil samples that are derived from background than those that overly the surface expression of the kimberlites. For the DO-18 field site, there are 14 soil samples derived from soils situated above the surface expression of the kimberlite, and 36 samples from background soils. At Kelvin, there are 21 soil samples above the surface expression of the kimberlite, and 34 samples from background soils. At the same time, the reviewer makes an excellent point, and the power of the approach definitely increases with more background analyses. This research is ongoing, but should also be prioritized by organizations like geological surveys that could build extensive background databases from which to compare microbial surface anomalies.

While the statistical analysis is standard, and I have no objections to it, I am not confident in the claims of unique bacterial community structure above subsurface kimberlite deposits: the bacterial species that undergo remarkable changes are a small fraction of the whole community.

This is true for individual species, but collectively, these organisms represent an appreciable fraction of the community. We have added this information explicitly into the main text (line 225 and 248-249), and as supplementary table (S7)

Field Site	Average summed % relative abundance across “ON” samples	Average summed % relative abundance across “OFF” samples	Average % Relative Abundance Total
DO-18	16.480 ± 8.384	7.250 ± 4.010	9.834 ± 6.904
Kelvin	7.768 ± 4.130	6.450 ± 4.433	6.953 ± 4.329

At the same time, I appreciate that the paper is well-written and easy to read.

Thank you.

11th Aug 23

Dear Dr Crowe,

Your manuscript titled "DNA sequencing, microbial indicators, and the discovery of buried mineral resources" has now been seen by our reviewers, whose comments appear below. In light of their advice we are delighted to say that we are happy, in principle, to publish a suitably revised version in Communications Earth & Environment under the open access CC BY license (Creative Commons Attribution v4.0 International License).

We therefore invite you to edit your manuscript to comply with our format requirements and to maximise the accessibility and therefore the impact of your work.

EDITORIAL REQUESTS:

We see that one of the co-authors is sadly listed as deceased. I'm afraid we must check with you that the author in question approved submission of the manuscript to Communications Earth & Environment, as this is one of our requirements for authorship (<https://www.nature.com/nature-portfolio/editorial-policies/authorship#authorship>). If they were not able to do so before they sadly passed away, we suggest their contribution is instead described in the acknowledgment section of the manuscript.

*****Please take care to match our formatting and policy requirements. We will check revised manuscript and return manuscripts that do not comply. Such requests will lead to delays. *****

SUBMISSION INFORMATION:

OPEN ACCESS:

Communications Earth & Environment is a fully open access journal. Articles are made freely accessible on publication under a [CC BY license](http://creativecommons.org/licenses/by/4.0) (Creative Commons Attribution 4.0 International License). This license allows maximum dissemination and re-use of open access materials and is preferred by many research funding bodies.

For further information about article processing charges, open access funding, and advice and support from Nature Research, please visit <https://www.nature.com/commsenv/article-processing-charges>

At acceptance, you will be provided with instructions for completing this CC BY license on behalf of all authors. This grants us the necessary permissions to publish your paper. Additionally, you will be asked to declare that all required third party permissions have been obtained, and to provide billing information in order to pay the article-processing charge (APC).

[link redacted]

Best regards,

Joe Aslin
Senior Editor,
Communications Earth & Environment
<https://www.nature.com/commsenv/>
Twitter: @CommsEarth

Sadia Ilyas
External Editor
Communications Earth & Environment

REVIEWERS' COMMENTS:

Reviewer #1 (Remarks to the Author):

The authors have thoughtfully considered and thoroughly addressed all reviewer comments on the first version of this manuscript. I do not have any additional comments and recommend the revised manuscript be accepted in its current form.

Reviewer #2 (Remarks to the Author):

The reviewer appreciates the efforts put in by the authors to revise their manuscript as per the review reports and provide a point-by-point response to each comment. I am in agreement with the response made to my queries and comments. Hence, I am happy to recommend the revised manuscript for possible publication in Communication Earth & Environment.

REVIEWERS' COMMENTS:

Reviewer #1 (Remarks to the Author):

The authors have thoughtfully considered and thoroughly addressed all reviewer comments on the first version of this manuscript. I do not have any additional comments and recommend the revised manuscript be accepted in its current form.

Thank you for reading the revised version and the positive comments

Reviewer #2 (Remarks to the Author):

The reviewer appreciates the efforts put in by the authors to revise their manuscript as per the review reports and provide a point-by-point response to each comment. I am in agreement with the response made to my queries and comments. Hence, I am happy to recommend the revised manuscript for possible publication in Communication Earth & Environment.

Thank you for reading the revised version and the positive comments